# Performance Enhancement of Flexible Polymer Triboelectric Generator through Polarization of the Embedded Ferroelectric Polymer Layer

Deahoon Park [1,2,†], Min Cheol Kim [1,†], Minje Kim [1], Pangun Park [1] and Junghyo Nah [1,*]

1   Department of Electrical Engineering, Chungnam National University, Daejeon 34134, Korea; dpark9277@utexas.edu (D.P.); mcheol1@o.cnu.ac.kr (M.C.K.); kmj93@cnu.ac.kr (M.K.); pgpark@cnu.ac.kr (P.P.)
2   Department of Electrical and Computer Engineering, The University of Texas at Austin, Austin, TX 78758, USA
*   Correspondence: jnah@cnu.ac.kr
†   These authors equally contributed.

**Abstract:** In this work, we report on a flexible triboelectric generator (TEG) with a multilayer polymer structure, consisting of a poly(vinylidene fluoride-co-trifluoroethylene) (PVDF-TrFE) layer sandwiched by polydimethylsiloxane (PDMS) layers for the performance enhancement of TEGs. We confirmed that the output performance of the TEG is closely dependent on the structure and polarization direction of the PVDF-TrFE layer. In addition, the PDMS layer serves as the electron trapping layer and suppresses the discharging of the surface charges, boosting the output performance. Furthermore, the polarized PVDF-TrFE layer in the preferred direction contributes to increasing the surface potential during the contact–separation motion. The interaction between these two polymer layers synergistically leads to the boosted output performance of TEGs. Specifically, the maximum peak-to-peak output voltage and current density of 420 V and 50 μA/cm² generated by the proposed architecture, representing approximately a fivefold improvement compared with the TEG with a single layer, even though the same friction layers were used for contact electrification.

**Keywords:** triboelectric generator; ferroelectric coupling effect; flexible polymers

## 1. Introduction

The triboelectric generator (TEG), which converts different mechanical stimuli into electrical energy through frictional electrification and electrostatic induction, has attracted great attention as a kinetic energy harvesting device, thanks to its relatively simple device structure and higher outputs compared with other energy harvesting devices [1–8]. Since triboelectric power generation is highly dependent on the triboelectric surface properties of the friction layers, proper treatment of the friction surface is the key to enhancing the performance of a TEG. To this end, several methods, such as surface patterning [9–12], chemical functionalization [13–15], and dielectric property modulation [16–18], have been introduced, which have demonstrated notable output performance enhancement of TEGs. Most of these approaches, however, focus only on the surface properties of the friction layers, but the role of the layer beneath the friction surface has been largely ignored. Recently, an interesting approach to embedding the charge storage inside or beneath the friction surface was tried, and significant output performance enhancement was achieved [19,20]. In addition, the improvement of the surface potential through the dipolar coupling effect by adopting ferroelectric materials was also recently reported [21–23]. These recent approaches exploit charge trapping or dipolar coupling of the sublayer to increase the surface charge density or potential of the triboelectric friction surface. However, the effective use of these methods is still unclear, and only a few results have been reported [24,25]. Furthermore, no attempts have been made to apply both methods simultaneously. Therefore,

further studies are necessary to develop the triboelectric friction layers that synergistically improve the performance of TEGs. In particular, only limited performance enhancement methods are available for the TEGs made of only flexible polymeric layers [26–29].

In this work, we report on a multi-layered (sandwich) polymer structure consisting of polydimethylsiloxane (PDMS) and polarized poly(vinylidene fluoride-co-trifluoroethylene) (PVDF-TrFE) layers to boost the surface charge density of the friction surfaces of a TEG. Here, the PDMS layer functions as a charge blocking layer and charge reservoir, providing charge trapping states at the interface between the PVDF-TrFE and the PDMS. In addition, the ferroelectric PVDF-TrFE layer plays dual roles as a negative triboelectric friction layer and surface charge modulation layer by polarizing the internal dipoles in a preferred direction. Our findings show that the outputs of TEGs can be greatly improved by forming a multilayered (sandwich) structure in the friction layer and properly polarizing the PVDF-TrFE layer. To determine the optimal multilayer structure composed of PVDF-TrFE and PDMS, we examined four different friction pairs of TEGs. Using these approaches, the TEG with the polarized sandwich structure generated the maximum peak-to-peak output voltage and current density of ~420 V and ~50 µA/cm$^2$, respectively, which were approximately values about five times higher than the TEG with a non-polarized single layer.

## 2. Materials and Methods

### 2.1. Materials

A silicone elastomer kit (Sylgard 184) and PVDF-TrFE powder (70:30 mol%) were purchased from Dow Corning (Midland, MI, USA) and Piezotech Arkema Corp (Colombes, France), respectively. *N,N*-Dimethylformamide (DMF) and acetone were purchased from Daejung chemicals and metals (Siheung, Korea). All chemicals were used without further purification steps.

### 2.2. Fabrication of Devices

A silicone elastomer with a 5:1 weight ratio (PDMS to curing agent) and a PVDF-TrFE solution (7 wt%) were used for the contact layers. For the PVDF-TrFE solution, PVDF-TrFE powder was dissolved in a co-solvent of DMF and acetone (1:1 volume ratio) at 70 °C while stirring for 2 h. All of the solutions used in this work were deposited via spin coating at 3000 rpm for 30 s, followed by a curing process on a hot plate at 90 °C for 20 min. The area of as-fabricated TEGs was kept at $2 \times 2$ cm$^2$.

### 2.3. Measurement and Characterization

The output voltage and current were measured by using an oscilloscope (Wavesurfer 3022, Lecroy, Chestnut Ridge, NY, USA) and a current preamplifier (SR570, Stanford research systems, Sunnyvale, CA, USA), respectively. The charge density was measured by using an electrometer (6514, Keithley, Solon, OH, USA). The custom-made vertical pressure machine was used to measure the triboelectric outputs. The Kelvin probe force microscopy (KPFM) was obtained using an XE7 (Park systems, Suwon, Korea). X-ray diffraction (XRD) analysis was performed using a D8 discover (Bruker, Billerica, MA, USA). The electrostatic voltmeter (ESVM) was measured using an ARS-S005 (SECOS, New Taipei City, Taiwan). The capacitor charging test was performed by using a high-precision digital multimeter (34401A, Agilent, Santa Clara, CA, USA). The polarization versus the electric field curves was measured by using a ferroelectric tester (RT66C, Radiant Tech, Albuquerque, NW, USA). Fourier-transform infrared spectroscopy (FTIR) analysis was conducted using Alpha (Bruker, Billerica, MA, USA). We must note that the active area, where vertical force was applied on the devices, was kept at $1 \times 1$ cm$^2$.

## 3. Results and Discussion

### 3.1. Role of the Buried Layer with the Performance

Schematic diagrams of the TEG fabrication process are illustrated in Figure 1a. A co-solvent of DMF and acetone was used for the synthesis of the PVDF-TrFE solution, and the mixture of the silicone elastomer and the curing agent was employed for the PDMS friction layer. Each solution was spin coated on a pre-cleaned indium tin oxide (ITO)-coated polyethylene terephthalate (PET) substrate. We should note that all the prepared samples were flexible. The detailed fabrication process of friction pairs is described in Section 2. To investigate the role of the buried layer on the performance of a TEG, the TEGs with different buried layers but identical surface friction pairs were prepared as shown in Figure 1b. We should note that the surface friction pairs of all the samples were kept as PVDF-TrFE and PDMS for a fair comparison. The output generation mechanism of the contact mode TEG is described in Figure S1 (Supplementary Materials). Figure 1c schematically shows the different charge trapping mechanisms of the buried PDMS layer and the buried PVDF-TrFE layer. PVDF-TrFE, which is on the relatively negative side of the triboelectric series, has a stronger electronegativity than PDMS, so when the two friction surfaces are contact-electrified, the electrons on the PDMS surface are transferred to the PVDF-TrFE surface [30]. Here, we note that PDMS is known to have deep trap states, while PVDF-TrFE has shallow trap states [31,32]. Thus, the electrons transferred to the PVDF-TrFE layer could be easily discharged to the ITO electrode layer if the PDMS blocking layer was not formed underneath (Figure 1c (left) and Figure 1b (iii)). On the other hand, when the electrons were transferred to the PVDF-TrFE surface with a buried PDMS layer, they were filling the deep states at the interface between the two layers, reducing the charge recombination and increasing the overall surface potential (Figure 1c (right) and Figure 1b (ii)) [32–34]. Figure 2a,b shows the measured output voltage and current density, respectively, of the as-fabricated TEGs with identical surface friction pairs but with different buried layers. For the output measurement, a pressure of 100 kPa normal to the contact surface was applied at 1 Hz for all the samples. Overall, the outputs of the TEGs with the buried layers were relatively higher than those of the TEGs with the single-layered friction pairs. Here, the buried layer served as an electron trapping layer. Thus, the TEGs with the buried layer could exhibit relatively high surface potentials during contact electrification (Figure 2c) and thus demonstrate relatively high output performances (Figure 2a,b). We must note that the output of a TEG with a buried PDMS layer (Figure 2a (ii)) was higher than that with a buried PDVF-TrFE layer (Figure 2a (iii)). The buried PDMS layer, which had a higher trap density compared with the PVDF-TrFE, alleviated the discharging of surface charges induced during the contact and separation motion. Therefore, the electrostatic surface potential of the buried PDMS could be higher than that of the buried PVDF-TrFE (Figure 2c (ii,iii)), leading to a similar trend in outputs (Figure 2a (ii,iii), Figure 2b (ii,iii)). Furthermore, as shown in Figure 2d, the output voltage of each TEG was monitored in real-time during the cyclic contact electrification. The results indicated that it took more time to reach the saturation voltage when the buried PDMS layer was adopted, due to the high trap density of the buried PDMS layer. Particularly, the highest output was obtained when the PVDF-TrFE layer was sandwiched between the PDMS layers (Figure 2a (iv)). Likewise, the electrostatic surface potential and voltage saturation time of each sample also represented a similar trend (Figure 2c,d).

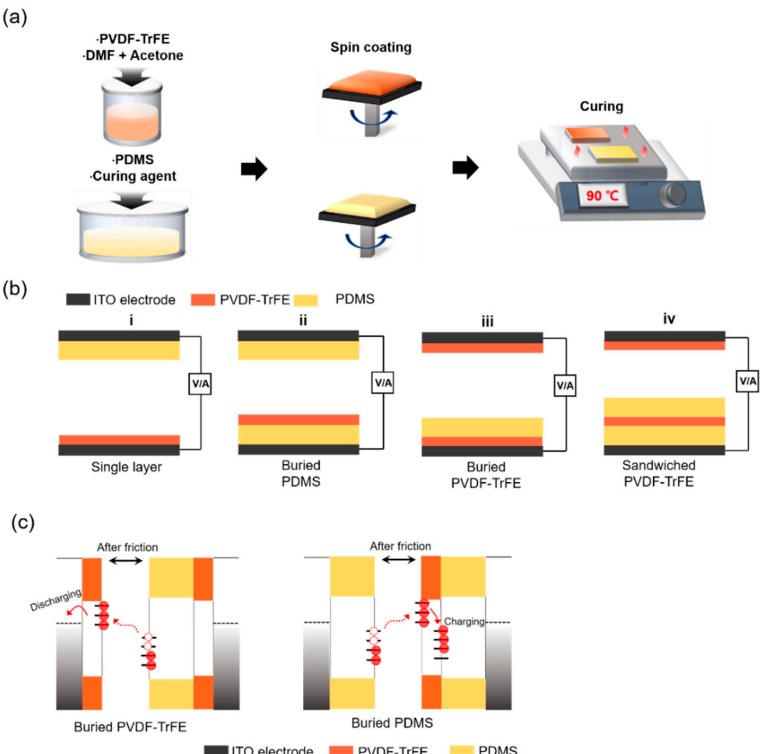

**Figure 1.** (**a**) Schematic fabrication process by the solution spin coating method. (**b**) Schematics showing four different contact friction pairs, consisting of polydimethylsiloxane (PDMS) and poly(vinylidene fluoride-co-trifluoroethylene) (PVDF-TrFE) layers. (**c**) Schematic diagram showing the charge trapping mechanism for two different buried layers: buried PVDF-TrFE (left) and buried PDMS (right).

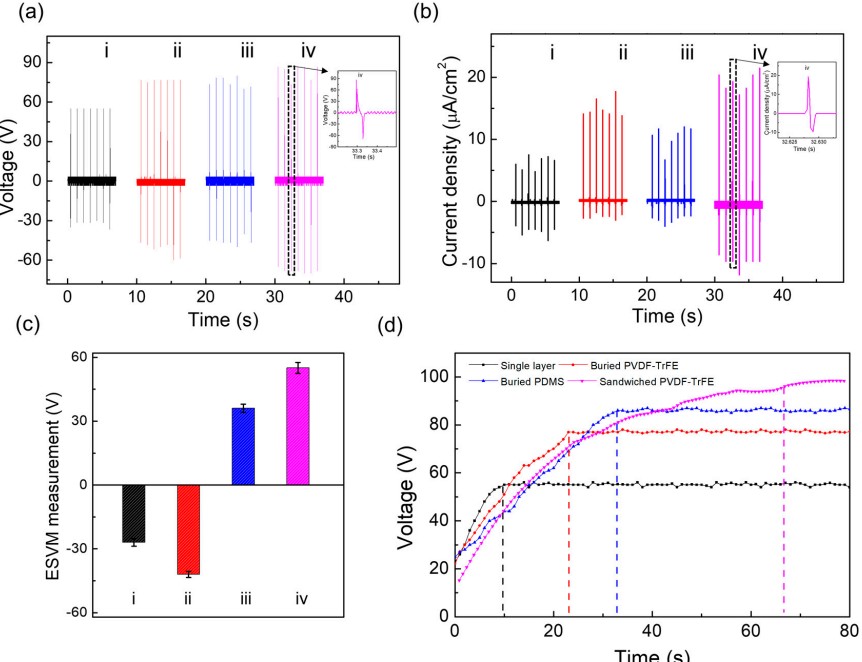

**Figure 2.** (**a**) Output voltage and (**b**) current density measured from each triboelectric generator (TEG). The inset is the detailed response of the voltage and current density. (**c**) Electrostatic voltmeter (ESVM) measurement of four TEGs after several contact–separation motions. (**d**) The saturated output voltage of four TEGs used in this study.

### 3.2. Ferroelectric Properties of PVDF-TrFE

Next, the polarization of the ferroelectric PVDF-TrFE layer and its dipolar coupling effect on the surface potential were investigated. Figure 3a schematically shows the polarization process of the ferroelectric PVDF-TrFE layer. In this work, we denoted negative polarization when the PVDF layer was polarized to have negative dipole centers on the friction surface and positive polarization when it was polarized to have positive dipole centers instead. For the polarization of the PVDF-TrFE layer, a high electric field of 24 kV/cm was applied between the electrodes while maintained at 90 °C for 6 h. Afterward, the polarization vs electric field (P-E) curve of the differently polarized PVDF-TrFE layers was measured. As shown in Figure 3b, the remanent polarization ($P_r$) increased after the polarization process, regardless of the polarization direction. In general, the spin-coated PVDF-TrFE film exhibited both $\alpha$- and $\beta$-phases, of which the $\beta$-phase exhibited a ferroelectric property. Thus, a relatively higher $\beta$-phase formation was desired to achieve the electric dipole alignment effect. To further investigate the augmentation of the $\beta$-phase through polarization, XRD and FTIR analyses were performed, as shown in Figure 3c and Figure S3 (Supplementary Materials), where both spectra indicated the increase of the $\beta$-phase and the suppression of the $\beta$-phase. These results show that the ferroelectric properties of PVDF-TrFE could be effectively improved by applying an appropriate polarization process. In addition, to investigate the influence of the polarization direction, the surface potential of each layer was measured by non-contact mode KPFM over an area of 25 $\mu m^2$. Figure 3d shows the surface potentials of the PVDF-TrFE layers after different polarizations: non-polarization, negative polarization and positive polarization. The results indicated that the polarization direction had an explicit influence on its surface potentials. The measured surface potentials of the non-, negative and positive polarizations were −330 mV, −410 mV and −260 mV, respectively. When the ferroelectric layer was electrically polarized in the negative or positive direction, the surface potential either increased or decreased compared with the non-polarized state, depending on the polarization direction.

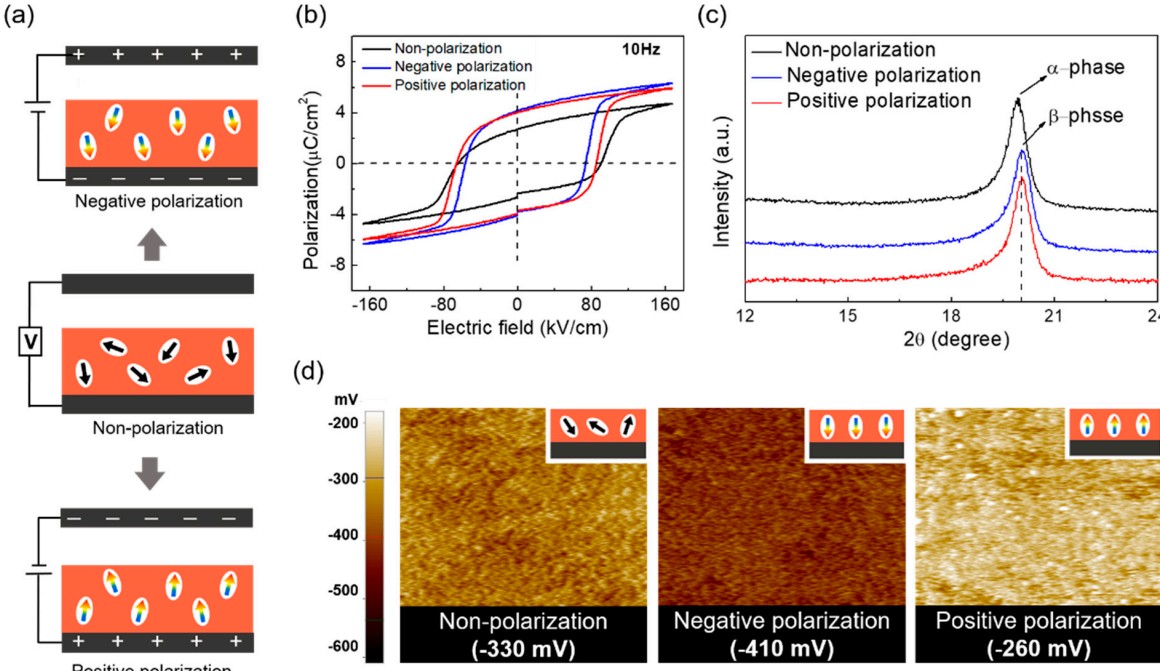

**Figure 3.** (**a**) Schematic diagrams of three different polarizations: negative polarization, non-polarization and positive polarization. (**b**) P-E curves of PVDF-TrFE after three different polarization processes. (**c**) X-ray diffraction (XRD) analysis of PVDF-TrFE, depending on the polarization direction. (**d**) Kelvin probe force microscopy (KPFM) images of the PVDF-TrFE surface for different polarizations scanned over an area of 25 $\mu m^2$.

### 3.3. Embedded Ferroelectric Polymer for the Performance Enhancement of TEGs

Figure 4a,b summarizes the peak-to-peak output voltage and current density measured from each TEG having different buried layers before or after the polarization process. Before polarization, the highest outputs were obtained from the TEG with PVDF-TrFE embedded in the PDMS layer. Even after polarization, the highest outputs were also observed from the same TEG structure. For all the TEGs with the buried layer, the output performance was further enhanced, regardless of the polarization direction, which was partially due to a permittivity increase after polarization. Using the results shown in Figure 4, both the triboelectrically positive and negative layers could be determined. As a negative contact friction layer for the TEG, either the PVDF-TrFE single layer or the PVDF-TrFE layer with buried PDMS layer could be used (Figure 1b (i,ii)). Here, the highest output was observed from the device with the negatively polarized PVDF-TrFE layer with buried PDMS. Negative polarization of the PVDF layer rendered the surface potential more negatively, and the induced surface charges were partially stored at the interface between the PVDF-TrFE layer and the PDMS layer. Next, a more obvious increase was achieved for the positive contact friction layer (Figure 1b (iv)). By positively polarizing the PVDF-TrFE layer embedded in PDMS, the output voltage was increased from 140 V to 425 V, which was approximately a threefold enhancement. Consistent results were also observed from the charge density measurement (Figure 4c). As shown in Figure 4d, by positively polarizing the PVDF-TrFE layer sandwiched between the PDMS layers, the PDMS friction surface could also be rendered more positively; thus, triboelectrically induced positive charges on the PDMS surface were screened less at the surface, contributing to increasing the surface potential and the outputs of the TEG. On the other hand, when the PVDF-TrFE layer was negatively polarized, the PDMS friction surface was terminated with negative charge centers; thus, triboelectrically induced positive charges were partially screened at the surface [19]. Therefore, it can be seen that the surface potential was greatly affected by the polarization direction of the dipoles in the embedded PVDF-TrFE layer. Here, the output of the TEG was rather decreased when the number of buried layers further increased, as shown in Figure S5 (Supplementary Materials). As the number of buried layers increased, the total thickness of the friction layer also increased, causing electrostatic induction losses and weakening the dipolar coupling effect. Thus, there exists an optimal number of layers to maximize the output performance.

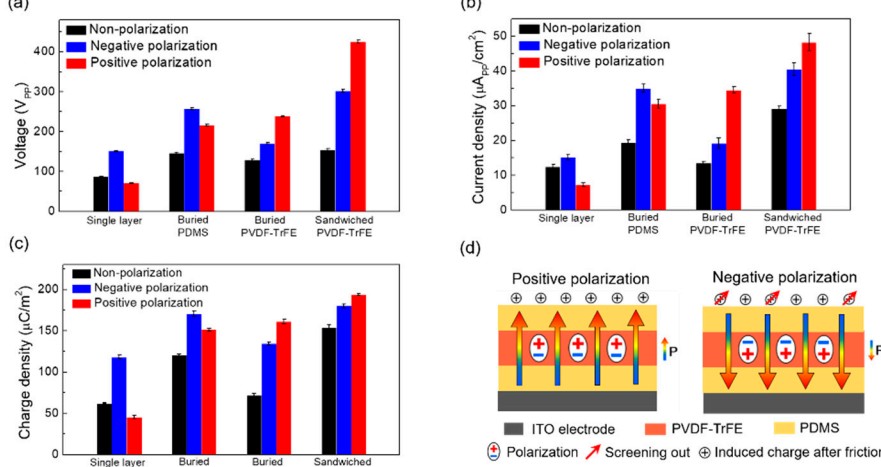

**Figure 4.** (**a**) Output voltage, (**b**) current density and (**c**) charge density measured from four different friction pairs, shown in Figure 1b, after non-, negative and positive polarization of the PVDF-TrFE layer. (**d**) Suppression of the triboelectrically induced positive surface charge screening effect by the positively polarized PVDF-TrFE layer (left) and screening of the triboelectrically induced positive surface charges by the negatively polarized PVDF-TrFE layer (right).

Lastly, we prepared a friction pair with buried layers to demonstrate the performance improvement of the TEG (Figure 5a). Here, as a negative friction layer, negatively polarized PVDF-TrFE with a buried PDMS layer was prepared. Since the PVDF-TrFE layer innately had a triboelectrically negative nature, this layer was used as a negative friction surface after negative polarization so that the negative center of the dipole was mainly distributed on the surface. During the cyclic contact friction, electrons on the friction surface could be partially stored at the interface with the PDMS layer, increasing the overall charge density of the layer. On the other hand, a positively polarized PVDF-TrFE-embedded PDMS layer was used as a positive friction layer. Here, since the PVDF-TrFE layer was triboelectrically negative compared with the PDMS layer, it was buried inside the PDMS layer to avoid the friction between the same surfaces and indirectly charge the PDMS surface positively through the polarized dipoles. We should note that both the PVDF-TrFE layers were polarized in a preferable direction. Figure 5b shows the output voltage and current density of the TEGs measured using the same friction pair with or without buried layers. The outputs measured after polarization of the PVDF-TrFE are shown next to each plot (red color). For the TEGs with the buried layers, the output voltage and current density of the TEG were clearly enhanced from 110 V to 360 V and from 15 $\mu$A/cm$^2$ to 47 $\mu$A/cm$^2$, respectively, after polarization of the PVDF-TrFE layer in a preferred direction. On the other hand, in the case of the TEG with single-layer friction pairs, an output voltage and current density of 100 V and 10 $\mu$A/cm$^2$ were measured, even after polarization. Therefore, the polarization effect could be boosted by forming a friction pair with buried layers. Next, the output voltage and current density of the TEG were also evaluated by varying the load resistances, as shown in Figure 5c. The output voltage increased with the connected load resistance while the current density decreased. A maximum output power density of ~3.3 mW/cm$^2$ was obtained by connecting a ~10 M$\Omega$ resistor (Figure 5d).

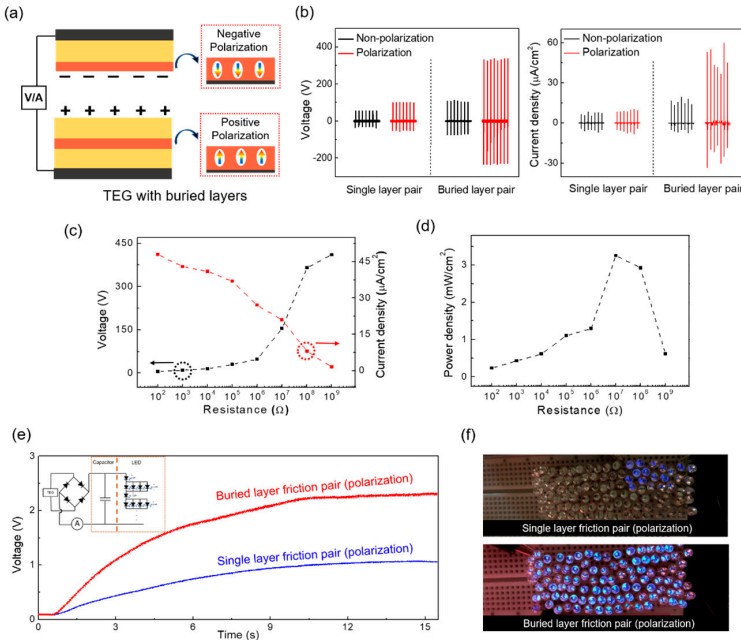

**Figure 5.** (**a**) Schematic representation of the TEG consisting of a friction pair with buried layers. (**b**) Output voltage and current density, measured using a single-layer friction pair and a friction pair with buried layers. The red colored lines show the outputs after polarization. (**c**) Dependence of the output voltage and current density on the load resistances. (**d**) Dependence of the power density on the load resistances. (**e**) Charging test using a capacitor (1 $\mu$F) for each contact friction pair. The inset is a schematic diagram of a full-wave bridge circuit connected to a capacitor or Light emitting diode (LED). (**f**) Instantaneous lighting of 20 (top) and 100 (bottom) blue LEDs during the contact–separation motions of two different contact friction pairs.

To explicitly demonstrate the performance of the TEGs, a capacitor (1 μF) was charged using each TEG. It can be clearly noticed that a capacitor could be quickly charged by adopting friction pairs with a buried PDMS layer and a polarized PVDF-TrFE layer (Figure 5e). In addition, while 20 LEDs can be turned on when using the TEG with single-layered friction pairs, the TEG with multilayered (sandwich) friction pairs could instantaneously turn on at least 100 LEDs (Figure 5f). Here, we constructed the load by the series and parallel connection of 100 commercial blue LEDs, as shown in Figure 5e (inset). Since the turn-on voltage of an LED is ~2.5 V, the total forward turn-on voltage of 100 LEDs connected as shown was ~100 V, which was lower than the open-circuit voltage of the TEG. By connecting this load to the output of the TEG, the output current through the circuit was measured as ~20 μA (Figure S6 in Supplementary Materials). Thus, the total resistance of 100 LEDs was ~5 MΩ, which was slightly smaller than the impedance at the maximum power density.

## 4. Conclusions

In this work, we reported the utilization of a ferroelectric PVDF-TrFE layer to improve the output performance of TEGs. Using four different TEG structures consisting of PVDF-TrFE and PDMS layers, we investigated the roles of buried polymer layers on the output performance of TEGs and proposed both positive and negative triboelectric friction layers. Specifically, by polarizing the ferroelectric PVDF-TrFE layer sandwiched between the PDMS layers in the preferred direction, the positive triboelectric friction surface charge density was greatly enhanced. On the other hand, a layer of PDMS serving as a charge reservoir was inserted between the polarized PVDF-TrFE friction surface and the electrode to increase the negative triboelectric friction surface charge density. Consequently, a maximum peak-to-peak output voltage and current density of 420 V and 50 μA/cm$^2$ were accomplished by the proposed architecture with the embedded polymer layers, demonstrating about a fivefold higher output when compared with the TEG having a non-polarized single friction layer. Until now, most of the studies have been focused on the properties of the friction surface to improve the surface charge density. However, in this work, we focused on the roles of the buried layer and revealed that the triboelectric surface charge density can also be greatly influenced by the layer buried under the friction surface. The approach proposed here provides an effective platform for the output enhancement of all polymer-based flexible TEGs, including a ferroelectric layer.

**Supplementary Materials:** The following are available online at https://www.mdpi.com/2076-3417/11/3/1284/s1. Figure S1: Schematic diagrams of the output generation mechanism of the contact mode TEG; Figure S2: The detailed voltage (a) and current (b) responses of Figure 2a, b; Figure S3: FTIR analysis of the PVDF-TrFE layers before and after polarization. The inset is the magnified plot of the a-phase; Figure S4: The raw data of the measured output voltage, current density and charge density of each TEG; Figure S5: (a) Schematic diagrams of TEGs consisting of four and five layers, consisting of PDMS and PVDF-TrFE, alternatively. (b) Measured output voltage and current density of the non-, negative-, and positive-polarized TEGs with four and five layers; Figure S6: The measured current through the 100 LEDs driven by the TEG.

**Author Contributions:** Conceptualization, J.N. and P.P.; experiment, D.P., M.C.K. and M.K.; formal analysis, D.P. and M.C.K.; writing—original draft preparation, D.P. and M.C.K.; writing—review and editing, J.N. and P.P.; funding acquisition, J.N. and P.P. All authors have read and agreed to the published version of the manuscript.

**Funding:** This research was supported by the Basic Science Research Program through the National Research Foundation of Korea (NRF-2019R1A2C1010384) and Korea Electric Power Corporation (R18XA06-04).

**Institutional Review Board Statement:** Not applicable.

**Informed Consent Statement:** Not applicable.

**Data Availability Statement:** Not applicable.

**Conflicts of Interest:** The authors declare no conflict of interest.

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
