# Peer review of "Performance Enhancement of Flexible Polymer Triboelectric Generator through Polarization of the Embedded Ferroelectric Polymer Layer"

_applsci, doi:10.3390/app11031284_

Round 1

Reviewer 1 Report

Triboelectric effect has gained much attention recently, however, it is important to evidence what is the novel part of the manuscript respect to the state art. Different triboelectric couples were investigated, also recently for TENG and triboelectric sensors. In particular PVDF and its co-polymers were amply investigated due to the triboelectric activity (10.1038/srep36409, 10.1016/j.nanoen.2019.104278). It is important to evidence what is the novel part of the proposed manuscript.

I suggest to modify a little bit Figure1 as initially supposed that PDMS and P(VDF-TrFe) were mixed together. Moreover, It should be important to have more details on the process reported in Figure 1 c. How authors verified that process.?

It would be interesting to have a mire detailed look of the voltage and current responses reported in figure 2a and b. What is the specific ESVM reported? it seems not reported in materials and methods. Units in figure 2c should be V not DV

I have some doubt concerning the poling process depicted in figure 3a. Why metal electrodes are not in touch with the layer? Moreover, what is the difference between positive and negative polarization introduced? Due to the fact that it results in very different triboelectric activity a more comprehensive discussion should be provided

What is the connection scheme of the LEDs, and a detailed discussion of their resistive load in comparison with those reported in figure 5 c and d would be helpful. I found discussion are very poor and there is also a lack of a “conclusion” which will be of help in summarizing obtained results.

Reviewer 2 Report

The manuscript reports on the effect of buried layer structures and polarization methods on the mechanical energy harvesting performance of TEG. The device with the optimal structure can generate the maximum peak-to-peak output voltage and current density of 420 V and 50 A/cm2 which are impressive. It is recommended for publication after the following issues are addressed:

  1. The full name of TEG should also be given in the abstract when it appears for the first time.
  2. The saturation voltage curve of the TEG device with the structure of PVDF-TrFE layer sandwiching between the PDMS layers should be added into Figure 2d.
  3. What are the thicknesses of PVDF-TrFE layer and PDMS layer? What is the effect of PVDF-TrFE layer thickness on device performance and ferroelectric property?
  4. What’s kind of polarization for the polarization curve in Figure 3b? The remnant polarization of PVDF-TrFE layer with negative polarization and positive polarization should be both illustrated in Figure 3b.

Round 2

Reviewer 1 Report

The manuscript has been moslty, improved.